# Unexpectedly High Prevalence of Breakfast Skipping in Low Body-Weight Middle-Aged Men: Results of the Kanagawa Investigation of Total Checkup Data from the National Data Base-7 (KITCHEN-7)

**DOI:** 10.3390/nu13010102

**Published:** 2020-12-30

**Authors:** Kei Nakajima, Ryoko Higuchi, Kaori Mizusawa

**Affiliations:** 1School of Nutrition and Dietetics, Faculty of Health and Social Services, Kanagawa University of Human Services, 1-10-1 Heisei-cho, Yokosuka, Kanagawa 238-8522, Japan; higuchi-nk3@kuhs.ac.jp (R.H.); mizusawa.hsp@kuhs.ac.jp (K.M.); 2Department of Endocrinology and Diabetes, Saitama Medical Center, Saitama Medical University, 1981 Kamoda, Kawagoe, Saitama 350-8550, Japan; 3Graduate School of Health Innovation, Kanagawa University of Human Services, Research Gate Building Tonomachi 2-A, 3-25-10 Tonomachi, Kawasaki, Kanagawa 210-0821, Japan

**Keywords:** breakfast skipping, irregular eating late-night dinner, late-night snack, low body weight, middle age, obesity, sex

## Abstract

Breakfast skipping (BS) has been considered to be associated with obesity, particularly among younger generations. However, few studies have addressed this issue in a middle-aged population considering sex and the conditions prior to breakfast. Therefore, we investigated clinical parameters, self-reported BS, late-night dinner (LND) eating, and late-night snacking (LNS) in ten body mass index (BMI) categories in a cross-sectional study of 892,578 non-diabetic people aged 40–74 years old who underwent a checkup. BS and LND were more prevalent in men (20.7% and 40.5%, respectively) than women (10.9% and 17.4%), whereas LNS was more prevalent in women (15.0%) than men (12.2%; all *p* < 0.0001). The overall prevalence of BS increased linearly with increasing BMI. However, when subjects were divided into men and women, the prevalence of BS showed a U-shaped relationship with BMI in men (n = 479,203). When male subjects were restricted to those in their 40s or those reporting LND, the prevalence of BS further increased, maintaining a U-shaped form. Logistic regression analysis also showed a U-shaped relationship in the adjusted odds ratios of BMI categories for BS in men and a J-shaped curve in women. In conclusion, our study revealed an unexpectedly high prevalence of BS in middle-aged low-body-weight men.

## 1. Introduction

Breakfast is the first meal of the day and plays a key role in overall eating behavior and metabolism of nutrients [1,2]. Breakfast skipping (BS) has been considered an unhealthy eating behavior that leads to unfavorable outcomes such as obesity, type 2 diabetes, metabolic syndrome, and higher mortality, particularly in younger age groups [1,2,3,4,5,6,7,8,9]. Furthermore, it has been suggested that the association between BS and type 2 diabetes may be partially mediated by body mass index (BMI) [5]. Although BS without overeating late in the day would reduce the total energy intake, which might logically contribute to weight loss, the mechanism underlying the association between BS and obesity remains poorly understood, and studies including clinical trials have produced conflicting results [2,10,11,12,13,14,15]. Late-night dinner (LND), eating before sleep, and taking a late-night snack (LNS) have also been reported to be associated with BS [2,6,16], indicating that other unhealthy eating behaviors prior to breakfast can influence physical conditions in the early morning, consequently contributing to the avoidance of breakfast.

In the middle-aged population, the proportions of cardiometabolic diseases such as type 2 diabetes, hypertension, and dyslipidemia are elevated and aggravated along with the peak prevalence of obesity [17,18]. Nevertheless, few studies of middle-aged populations have addressed the issue of BS and obesity while considering confounding conditions including LND and LNS. In addition, sex-related differences may exist in the association between BS and obesity, which have not been fully considered in previous studies [1,2,3,4].

We therefore investigated the association between BS and a wide range of classifications in body weight considering confounding factors in a cross-sectional study using a large population-based dataset of 892,578 non-diabetic people (479,203 men and 413,375 women) aged 40–74 years old who underwent a regular checkup.

## 2. Materials and Methods

### 2.1. Study Design and Subjects

We performed a composite multidisciplinary study including secondary use of annual health checkup data in Japan (Kanagawa Investigation of the Total Checkup Data from the National Database; KITCHEN) to investigate the clinical factors primarily associated with cardiometabolic diseases. Details of the study concept and design have been published elsewhere [19]. The present study included all individuals who underwent these specific health checkups and were living in Kanagawa Prefecture. The study protocol was approved by the Ethics Committee of Kanagawa University of Human Services (10–43) and the Ministry of Health, Labor, and Welfare of Japan (No. 121).

We initially reviewed the data collected from 1,819,173 people aged 40–74 years who attended health checkups between April 2013 and March 2014. People who underwent pharmacotherapy for diabetes, regardless of type 2, type 1, or others, were excluded (n = 113,097; 6.2%) because pharmacotherapy for diabetes can unintentionally influence body weight. In addition, people who had HbA1c of ≥6.5% were also excluded because they were likely to be following a specific diet, either by themselves or in consultation with dieticians. After further exclusion of subjects with incomplete data, 892,578 subjects remained for the study analysis (479,203 men and 413,375 women).

We received digitally recorded anonymous data from the Ministry of Health, Labor, and Welfare of Japan in 2017, as part of its nationwide program involving the provision of medical data to third parties [20]. To protect against the identification of specific individuals, their ages had been categorized as 40–44, 45–49, 50–54, 55–59, 60–64, 65–69, or 70–74 years. In this study, however, to evaluate subject age as a single numeric value, we transformed the age groups into substituted ages (s-age), corresponding to the median for each age group (42, 47, 52, 57, 62, 67, and 72 years, respectively).

### 2.2. Measurements

Anthropometric and laboratory measurements were conducted in the morning following an overnight fast. Body weight, waist circumference at the navel level, and height were objectively measured by trained institutional staff members. BMI was calculated as mass (kg) divided by the square of height (m^2^). Subjects were classified into 10 BMI groups: ≤ 16.9, 17–18.9, 19–20.9, 21–22.9, 23–24.9, 25–26.9, 27–28.9, 29–30.9, 31–32.9, and ≥ 33.0 kg/m^2^. Apart from this classification, we divided all subjects into two groups: Non-obese (BMI < 25.0 kg/m^2^) and overweight/obese (BMI ≥ 25.0 kg/m^2^) to investigate a rough association of BS with obesity. Furthermore, subjects were classified into 10 waist circumference (WC) groups: ≤ 64.9, 65.0–69.9, 70.0–74.9, 75.0–79.9, 80.0–84.9, 85.0–89.9, 90.0–94.9, 95.0–99.9, 100.0–109.9, and ≥110.0 cm, to compare the results obtained with two obese indices of BMI and WC. The measurements of blood pressure and blood was regularly standardized using both internal standards with available traceability and external standards by third parties [19]. Laboratory measurements were performed automatically using standard methods.

Questions for detecting unhealthy eating habits were developed by the Japanese Ministry of Health, Labor and Welfare in 2008 [19,21]. Habitual BS, LND, and LNS were determined on the basis of a positive response to the question: “Do you skip breakfast at least three times per week?”, “Do you eat dinner within 2 h before bedtime at least three times per week?”, and “Do you eat snacks after dinner more than three times per week?”, respectively. In this study, LND did not necessarily reflect that dinner was eaten at a particularly late time of the day (e.g., around midnight), but rather that dinner was taken shortly before an individual’s bedtime.

### 2.3. Statistical Analysis

Data were expressed as means ± SD or medians (interquartile range). Differences in continuous and categorical variables were evaluated by analysis of variance and the χ^2^ test, respectively. Trends in the prevalence of BS across the increasing BMI strata were evaluated by Cochran-Armitage tests. A logistic regression model was used to evaluate the associations between BMI/WC categories and BS, with adjustment for potential confounding factors (age, sex, LND, LNS, pharmacotherapy for hypertension or dyslipidemia, smoking, the amount of alcohol consumption, and habitual exercise), and yielded adjusted odds ratios (ORs) and 95% confidence intervals (CIs). As in previous studies, BMI of 21.0 to 22.9 kg/m^2^ was considered the provisional reference for the BMI category [16]. All statistical analyses were performed using SAS-Enterprise Guide (SAS-EG 7.1) in SAS software, version 9.4 (SAS Institute, Cary, NC, USA). Values of *p <* 0.05 were considered to represent statistical significance. When differences in BS prevalence between two selected BMI groups among ten groups were evaluated by the χ^2^ test, values of *p <* 0.001 were considered to represent statistical significance, on the basis of the Bonferroni test.

## 3. Results

Table 1 shows the characteristics of the subjects grouped into men and women. Significant statistical differences between men and women were observed in all continuous and categorical valuables (all *p*-values < 0.0001), probably due to the size of the dataset. The prevalence of underweight (BMI < 18.5 kg/m^2^) was higher in women (13.9%) than men (3.0%); the actual number of underweight subjects in women (n = 57,406) was four times that of men (n = 14,185). The prevalence of BS and LND were both higher in men (20.7% and 40.5%, respectively) than in women (10.9% and 17.4%), whereas for LNS there was a smaller difference between women (15.0%) and men (12.2%).

In the cohort as a whole (both men and women), the prevalence of BS increased linearly across the increasing BMI strata (Figure 1a, *p <* 0.0001, Cochran-Armitage test). However, when subjects were divided into men and women, the prevalence of BS showed a U-shaped relationship against BMI in men. The prevalence of BS in the lowest BMI category, ≤ 16.9 kg/m^2^, (25.8%) was significantly higher than that in the reference BMI group of 21.0–22.9 kg/m^2^ (19.4%, *p <* 0.0001), whereas no difference was observed between the lowest and highest BMI groups: ≤16.9 kg/m^2^ and ≥33.0 kg/m^2^ (27.2%) (*p* = 0.15). In contrast, in women the prevalence of BS showed a slight J-shaped relationship with increasing BMI, where no significant difference was observed between the lowest BMI group (11.0%) and the reference BMI group (10.3%, *p* = 0.21). When male subjects were restricted to those in their 40s or those with LND or LNS (Figure 1b and Figure 2a,b), the prevalence of BS increased, maintaining the U-shaped relationship. By contrast, as shown in Figure 3, the prevalence of LND and LNS each increased linearly with increasing BMI, regardless of men and women, indicating that LND and LNS were less prevalent in men with low BMI.

A logistic regression analysis showed a linear relationship between crude ORs of BMI categories for BS in the set of all subjects (Figure 4a). However, when subjects were divided into men and women, the relationship was transformed into U-shaped relationships in men, which was further transformed into a blunt left-right inverted J-shaped relationship after adjustment for confounding factors (Figure 4b). In women, a slight J-shaped relationship was observed between crude ORs and BMI categories (Figure 4a), which was largely unchanged even after adjustment for confounding factors (Figure 4b). When men and women were combined and adjusted ORs were calculated, a slight J-shaped relationship was observed, similar to that in women alone (Figure 4b). Appendix A shows concrete values of odds ratios and 95% confidence intervals.

Although the data are not shown in the form of a Figure, when the ten BMI categories were replaced with dichotomized BMI groups (<25.0 and ≥25.0 kg/m^2^) in the final model that adjusted for confounding factors, BMI of ≥25.0 kg/m^2^ was significantly but modestly associated with BS in both men (OR (95% CIs), 1.04 (1.02–1.05), *p <* 0.0001), and women (1.27 (1.24–1.31), *p* < 0.0001).

When 10 BMI categories were replaced with 10 WC categories, similar results were observed, although U-shaped association became rather a J-shaped association in men (Appendix A).

## 4. Discussion

In recent decades, many cross-sectional studies have shown positive associations between BS and obesity, whereas some cross-sectional and cohort studies and randomized controlled trials have not [2,10,11,12,13,14,15]. In spite of our study being cross-sectional in nature, its use of a very large healthcare database allowed classification into multiple BMI categories, showing that compared with reference weight, both excess and low body weights were associated with BS in men, as reflected by a U- or J-shaped relationship, which was more clearly evident in men regardless of their age, LND, or LNS. By contrast, the association between low body weight and BS was not as clearly observed in women, as reflected in a slight J-shaped relationship, which in turn suggests the possibility that the association between BS and obesity may be more straightforward in middle-aged women. Noteworthy, the current findings above were not largely altered when the BMI category was replaced with the WC category. In addition, LND and LNS may be critical factors to be addressed, equal to or beyond BS, in terms of obesity both in men and women, because of the linear relationship between these unhealthy eating habits and BMI. Notably, because the proportion of subjects reporting BS increased in parallel with those reporting LND and LNS, LND and LNS may be associated with increased BS independently of BMI (Figure 2a,b).

The current results might not have been determined until we were able to use a very large database including healthcare data from 890,000 people, which allowed us to conduct detailed classification including BMI less than 17.0 kg/m^2^, a criterion for moderate and severe thinness [22], and to consider sex-related differences, age, and confounding factors. Because underweight men are generally less prevalent, as in this study (3.0%, Table 1), small and moderately large studies have been unable to conduct detailed analyses relating BS to multiple BMI categories. If we had analyzed no more than the simple association between obesity and BS, which was also confirmed in this study, we would claim that BS is associated with obesity in both men and women as reported in previous studies [1,2,4,5,6,7,9], but we would also overlook the current findings of unexpectedly high prevalence of BS in low-body-weight men.

In our previous study of 60,000 people living in a different region from that currently investigated [16], the results showed a J-shaped relationship between the prevalence of BS and BMI in a population-based study enrolling men and women. In that study, the proportion of men was higher (62.7%) than in the current study (53.7%), which might strengthen the effect of the specific characteristic among low-body-weight men, thereby showing a mild or moderate J-shaped relationship of BS against BMI, even when the analysis was conducted for the entire cohort. Unfortunately, due to its relatively small sample size compared with the current study, that study did not analyze the sex-related difference in the results.

To our knowledge, this study is the first to show a higher prevalence of BS in low-body-weight middle-aged men than among corresponding reference-body-weight men, and a slight J-shaped association in women. However, the direction of causality between BS and obesity was unknown due to the cross-sectional study design, and we are unable to conclude whether BS contributes to low body weight in the middle-aged men, or whether such men frequently skip breakfast. In addition, the reasons for the sex-related differences in the associations between BS and BMI remain unclear.

A plausible explanation for the high prevalence of BS in low-body-weight men is that BS would theoretically reduce the total energy intake for the day, unless overeating and compensatory intake were to follow the skipped meal. This would result in BS acting as an intermittent fasting diet [2,23], which might be feasible in some proportion of lean men [24].

Unfortunately, this study could not determine whether total energy consumption is reduced or increased in breakfast skippers, which is critical information to address the issue. Some studies have shown higher energy intake among breakfast skippers [1,2,25], whereas others have not [11,23,26,27], suggesting that there may be several types of breakfast skippers according to high/low levels of energy intake and expenditure. Further study is needed to elucidate whether BS is a cause of low body weight in middle-aged men. Clinical trials of breakfast consumption in low-body-weight men should be undertaken to confirm the current results.

Meanwhile, appetite-regulating substances including leptin, neuropeptide Y, ghrelin, peptide YY, and orexin may contribute to the sex-related differences in the association between BS and BMI, as several human and animal studies have reported the possibility of sex-related differences in these substances [28,29,30,31]. Consistently, eating disorders such as anorexia nervosa and binge eating disorder have been reported to be more prevalent in women than in men [32,33]. However, it is unknown whether BS in low-body-weight men is attributable to such eating disorders or impaired mental health, although BS may be associated with depressive symptoms and mood disorders [34,35,36]. This issue should be elucidated in further study investigating mental health.

## 5. Limitation

There are several limitations in addition to those mentioned above. First, as there are no universally agreed definitions of habitual BS, LND, and LNS, the presence or absence of these unhealthy eating habits depends on the individual’s assessment, although a common set of questions by the Ministry of Health, Labor, and Welfare of Japan were provided to evaluate these eating habits [21]. Second, no investigation of meal contents, for instance using a Food Frequency Questionnaire, was conducted in this study, which hampered investigation of the relationship among consumed energy and nutrients, BS, and BMI. However, a scientific statement from the American Heart Association has mentioned the possibility that the association between BS and obesity may be independent of differences in diet quality between people with BS and those without [2]. Third, the contents in pharmacotherapy for hypertension or dyslipidemia were unknown in this study, which may influence the results. Finally, in Japan as in the rest of Asia, underweight people are much more prevalent than in western countries [37,38]. In addition, individuals with diabetes were excluded in this study. Therefore, our results may be less applicable to other countries with higher BMIs and other studies including people with diabetes.

## 6. Conclusions

Our study confirmed the longstanding belief that the overall prevalence of BS increases linearly and significantly with increasing BMI. However, the current results also revealed an unexpectedly higher prevalence of BS in middle-aged low-body-weight men compared with reference-weight men, which may be increased by the concurrence of LND and LNS. By contrast, the association between BS and obesity may be more straightforward in middle-aged women.

## Figures and Tables

**Figure 1 nutrients-13-00102-f001:**
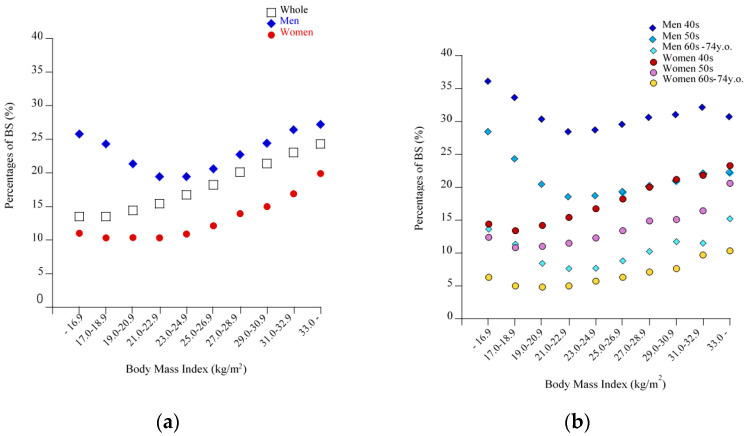
Percentages of breakfast skipping (BS) according to BMI categories, sex, and three age groups. (**a**) All subjects, men, and women. (**b**) Men in their 40s, 50s, and 60–74 years old, and women in their 40s, 50s, and 60–74 years old. Each symbol indicates the percentage for that group.

**Figure 2 nutrients-13-00102-f002:**
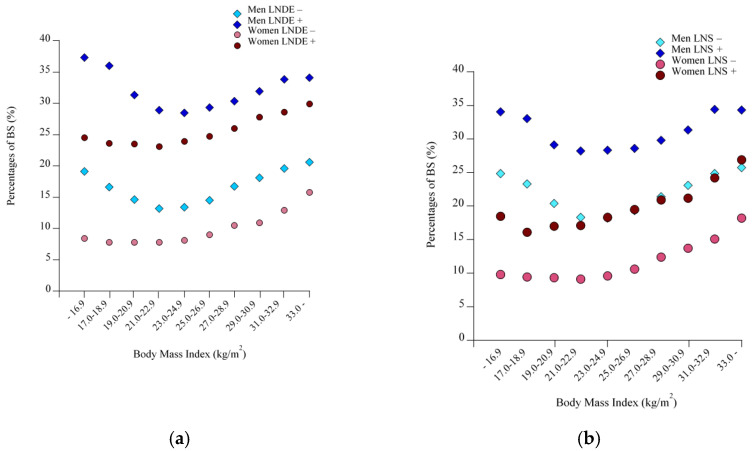
Percentages of breakfast skipping (BS) according to BMI categories, sex, and LND or LNS. (**a**) Men with or without LND, and women with or without LND. (**b**) Men with or without LNS, and women with or without LNS. Each symbol indicates the percentage for that group. BS: Breakfast skipping, LND: Late-night dinner, LNS: Late-night snack.

**Figure 3 nutrients-13-00102-f003:**
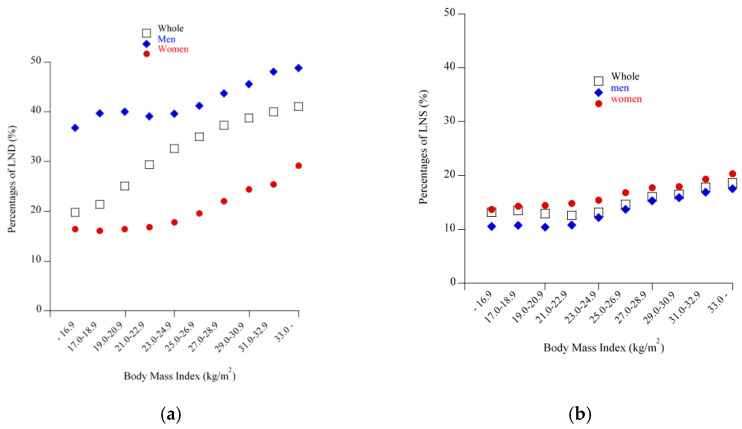
Percentages of late-night dinner (LND) and late-night snack (LNS) according to BMI categories and sex. (**a**) LND: All subjects, men, and women. (**b**) LNS: All subjects, men, and women. Each symbol indicates the percentage for that group.

**Figure 4 nutrients-13-00102-f004:**
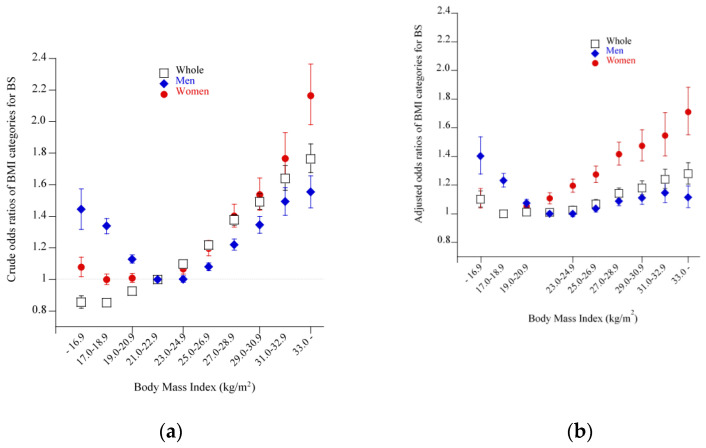
Odds ratios and 95% confidence intervals of BMI categories for breakfast skipping (BS). (**a**) Crude odds ratios and 95% confidence intervals for all subjects and for men and women separately. (**b**) Odds ratios and 95% confidence intervals for men and women after adjustment for age, smoking, pharmacotherapy for hypertension, pharmacotherapy for dyslipidemia, history of cardiovascular disease, moderate to heavy consumption of alcohol (≥46 g ethanol per day), habitual exercise, LND, LNS, and sex (in the case of all subjects). The symbols and bars express odds ratios and 95% confidence intervals. BS: Breakfast skipping, LND: Late-night dinner, LNS: Late-night snack.

**Table 1 nutrients-13-00102-t001:** Characteristics of subjects.

	Men	Women
N (% of total)	479,203 (53.7)	413,375 (46.3)
s-Age (years)	53.4 ± 9.5	54.9 ± 10.2
BMI (kg/m^2^)	23.7 ± 3.2	21.7 ± 3.4
Underweight Subjects, n (%) ^a^	14,185 (3.0)	57,406 (13.9)
Systolic blood pressure (mmHg)	124 ± 16.0	118 ± 17.5
High-density lipoprotein-cholesterol (mg/dL)	59.2 ± 15.3	72.2 ± 16.6
Triglyceride, IQR (mg/dL)	102 (72–148)	73 (54-101)
HbA1c (%)	5.5 ± 0.3	5.4 ± 0.3
Pharmacotherapy for hypertension, n (%)	85,858 (17.9)	52,328 (12.7)
Pharmacotherapy for dyslipidemia, n (%)	45,264 (9.5)	47,200 (11.4)
Cardiovascular disease, n (%)	15,835 (3.3)	7804 (1.9)
Current smoking, n (%)	147,013 (30.7)	37,348 (9.0)
Habitual exercise, n (%) ^b^	142,599 (29.8)	118,838 (28.8)
Moderate to heavy alcohol drinking, n (%) ^c^	11,6977 (24.4)	21,910 (5.3)
Breakfast skipping, n (%) ^d^	99.0 (20.7)	45,201 (10.9)
Late night dinner, n (%) ^e^	193,935 (40.5)	71,911 (17.4)
Late night snack, n (%) ^f^	58,525 (12.2)	62,144 (15.0)

Differences in continuous and categorical variables were evaluated by analysis of variance and the χ^2^ test, respectively (all *p*-values < 0.0001). ^a^ Defined as body mass index of < 18.5 kg/m^2^. ^b^ Defined as habitual exercise to a light sweat for over 30 min per session twice weekly. ^c^ Defined as daily alcohol consumption of 46 g ethanol or more. ^d^ Defined as skipping breakfast at least three times per week. ^e^ Defined as eating dinner within two hours before bedtime at least three times per week. ^f^ Defined as eating snacks after dinner more than three times per week. s-Age: Substituted age; BMI: Body mass index; IQR: Interquartile ratio.

## Data Availability

Please refer to suggested Data Availability Statements in section “MDPI Research Data Policies” at https://www.mdpi.com/ethics.

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
