# Peer review of "Unexpectedly High Prevalence of Breakfast Skipping in Low Body-Weight Middle-Aged Men: Results of the Kanagawa Investigation of Total Checkup Data from the National Data Base-7 (KITCHEN-7)"

_nutrients, 2020, doi:10.3390/nu13010102_

Round 1

Reviewer 1 Report

The authors proposed that breakfast skipping is prevalent in middle aged men taking into account the factors such as late night dinner and late night snacking. The authors conclude that high prevalence of BS in middle aged low weight men.

  1. What is the purpose of this study and how is it relevant to the scientific community. The authors have said that BS is prevalent in middle age men. If so, how is it going to affect the quality of the life or what consequences does it have.
  2. Just a study involving BMI is not conclusive across the population without knowing the outcome of the study.
  3. The study exclude diabetic patients, but not other metabolic diseases or taking treatment for any of the diseases. How are the authors sure that this does not affect the outcome of the study.
  4. BMI is influenced by many factors. Taking this parameter alone and associating it with breakfast skipping does not make any sense.
  5. The methods does not describe anything about how the metabolic parameters were done.
  6. Why did the study classify the pharmacotherapy for hypertension and dyslipidemia as one category when different people might have been given different drugs that would affect the individuals differently?
  7. Current smoking. What was the baseline for the study. The study classifies light and heavy smokers into one.
  8. The same applies for habitual exercise. Those who are on exercise regime are prone to follow a particular diet. How do the authors know that it is not influencing the outcome of the study.
  9. The study has not scientific importance to the community, not all parameter are taken into account in the study and importantly there is no conclusion why this was done.

Author Response

Thank you for reviewing our manuscript. We appreciate the reviewer.

  1. What is the purpose of this study and how is it relevant to the scientific community. The authors have said that BS is prevalent in middle age men. If so, how is it going to affect the quality of the life or what consequences does it have.

Response.

Low body weight has been shown to be associated with health problems including high mortality and fail. Meanwhile, breakfast skipping, a representative unhealthy eating behavior, has traditionally been believed to be prevalent in younger obese people.

However, the association between low body weight and breakfast skipping is unknown, particularly in a middle-aged population. Consequently, we found an association between low body weight and breakfast skipping in low body weight middle-aged men.

Then, current results suggest that breakfast skipping might contribute to the health problems frequently observed in low body weight people, which deserves further study in the future.

We added the next sentence in the introduction section.

“Although BS without overeating late in the day would reduce the total energy intake, which might logically contribute to weight loss,”

  1. Just a study involving BMI is not conclusive across the population without knowing the outcome of the study.

Response.

We agree with the comment and reanalyzed the data in terms of waist circumference, a surrogate marker for abdominal adiposity. As a result, we found similar results, which were shown in supplementary data.

We added the next sentence in the methods section.

“Furthermore, subjects were also classified into 10 waist circumference (WC) groups: ≤ 64.9, 65.0–69.9, 70.0–74.9, 75.0–79.9, 80.0–84.9, 85.0–89.9, 90.0–94.9, 95.0–99.9, 100.0–109.9, and ≥ 110.0 cm, to compare the results obtained with two obese indices of BMI and WC.”

In the results section.

“When 10 BMI categories were replaced with 10 WC categories, similar results were observed, although U-shaped association became rather a J-shaped association in men (Supplementary Table 2).

  1. The study exclude diabetic patients, but not other metabolic diseases or taking treatment for any of the diseases. How are the authors sure that this does not affect the outcome of the study.

Response.

We agree with the comment. However, the prevalence of subjects who had diabetes is small in low body weight people in general and in our previous study (less than 2%, Nakajima K, et al. Endocr Res. 2016) compared with the prevalence of those with breakfast skipping (20-25% in current study). Then, if we enrolled subjects with diabetes in our analysis, we speculate that the inclusion would not largely affect the results, although the actual outcomes should be confirmed in further study.

We added this issue in the limitation section as follows.

“In addition, individuals with diabetes were excluded in this study. Therefore, our results may be less applicable to other countries with higher BMIs and other studies including people with diabetes.”

  1. BMI is influenced by many factors. Taking this parameter alone and associating it with breakfast skipping does not make any sense.

Response.

We agree with the comment and analyzed the data in terms of waist circumference, a surrogate marker of abdominal adiposity. As a result, we found similar results, which were shown in supplementary data.

In the results section, we added the next sentence.

“When 10 BMI categories were replaced with 10 WC categories, similar results were observed, although U-shaped association became rather a J-shaped association in men (Supplementary Table 2).

  1. The methods does not describe anything about how the metabolic parameters were done.

Response.

We agree with the comment and added the next sentences in the measurements of methods section.

“Body weight, waist circumference at the navel level, and height were objectively measured by trained institutional staff members.”

“The measurements of blood pressure and blood was regularly standardized using both internal standards with available traceability and external standards by third parties [19]”

  1. Why did the study classify the pharmacotherapy for hypertension and dyslipidemia as one category when different people might have been given different drugs that would affect the individuals differently?

Response.

We agree with the comment. Unfortunately, however, the information for the contents of hypertension and dyslipidemia are unavailable in this study. We have only the data of absence or presence of pharmacotherapy for hypertension or dyslipidemia (Table 1).

We added the next sentences in the limitation section.

“Third, the contents in pharmacotherapy for hypertension or dyslipidemia were unknown in this study, which may influence the results.”

  1. Current smoking. What was the baseline for the study. The study classifies light and heavy smokers into one.

Response.

We agree with the comment. However, the data of current smoking status (yes or no) is only available.

  1. The same applies for habitual exercise. Those who are on exercise regime are prone to follow a particular diet. How do the authors know that it is not influencing the outcome of the study.

Response.

We agree with the comment. We have realized the limitation in the data of diet. We described this issue in the limitation section as follows.

“Second, no investigation of meal contents, for instance using a Food Frequency Questionnaire, was conducted in this study, which hampered investigation of the relationship among consumed energy and nutrients, BS, and BMI.”

  1. The study has not scientific importance to the community, not all parameter are taken into account in the study and importantly there is no conclusion why this was done.

Response.

We agree with the comment. We would like to consider more relevant parameter such as diet and physical activity, whereas such parameters are unavailable in this study. This is one reason why we submit our article as a communication instead of original article.

Reviewer 2 Report

The authors report on a secondary analysis of annual health checkup data to investigate the association between Breakfast Skipping (BS) and BMI, late-night dinner, late-night snacking and a wide range of confounding factors. This is a cross-sectional study using a large sample of women and men between 40-74 years in Japan. The authors are addressing a worthwhile research question and present interesting findings that further our knowledge. The manuscript is well written and easy to read and follow. Altogether, I have some minor comments for the authors.

Specific comments:

Introduction: The authors discuss mainly studies from Asia. There is some work available, which would be relevant, e.g.:

  • Keski-Rahkonen, A., Kaprio, J., Rissanen, A., Virkkunen M., Rose, R.J. (2003). Breakfast skipping and health-compromising behaviors in adolescents and adults. European Journal of Clinical Nutrition, 57, 842-853.
  • Ballon, A., Neuenschwander, M., Schlesinger, S. (2019). Breakfast Skipping Is Associated with Increased Risk of Type 2 Diabetes among Adults: A Systematic Review and Meta-Analysis of Prospective Cohort Studies. The Journal of Nutrition, 149(1), 106-113.

Materials and Methods:

Page 2, lines 72-76: I do not clearly understand how you dealt the single numerical values in the logistic regression analysis. Did you use dummy coding?

Page 2, line 80: I think the BMI group 19-20.8 should be 19-20.9

Page 2, line 81-82: why did you divide subjects into two groups (non-obese and overweight/obese), and not into three (underweight, normal weight, overweight/obese)?

Results:

Page 3, line 113: it would be better to compare the percentages of underweight women and men

Table 1: I think the superscript c is not correct in the line „Underweight Subjects“

Page 4, lines 139-141: I think, this text should be deleted

Presentation of results: I very much like the presentation of the results in the form of graphs. It makes the results visible and the paper easy to read. However, I think it would be good to provide tables with the results of the logistic regression analyses as supplementary material.

Discussion: The possible role of eating disorders should be discussed in more details. It is well-known that eating disorders are associated with breakfast skipping. Howerver, the authors only mention in the last sentence that eating disorders are more prevalent in women than in men.

Author Response

  1. Introduction: The authors discuss mainly studies from Asia. There is some work available, which would be relevant, e.g.:Keski-Rahkonen, A., Kaprio, J., Rissanen, A., Virkkunen M., Rose, R.J. (2003). Breakfast skipping and health-compromising behaviors in adolescents and adults. European Journal of Clinical Nutrition, 57, 842-853.

Ballon, A., Neuenschwander, M., Schlesinger, S. (2019). Breakfast Skipping Is Associated with Increased Risk of Type 2 Diabetes among Adults: A Systematic Review and Meta-Analysis of Prospective Cohort Studies. The Journal of Nutrition, 149(1), 106-113.

Response.

Thank you for useful papers. We read the articles with interest.

We added the second paper by Ballon et al. as a new reference (no. 5) with the next sentence in the introduction section.

“Furthermore, it has been suggested that the association between BS and type 2 diabetes may be partially mediated by body mass index (BMI) [5].”

In the references section,

  1. Ballon A, Neuenschwander M, Schlesinger S. Breakfast Skipping Is Associated with Increased Risk of Type 2 Diabetes among Adults: A Systematic Review and Meta-Analysis of Prospective Cohort Studies. J Nutr. 2019;149:106-113.

  1. Materials and Methods:

Page 2, lines 72-76: I do not clearly understand how you dealt the single numerical values in the logistic regression analysis. Did you use dummy coding?

Response.

We did not use dummy coding for the continuous variables. In the logistic regression analysis, continuous variables are treated as they are.

  1. Page 2, line 80: I think the BMI group 19-20.8 should be 19-20.9

Response.

That is our mistake. We corrected 19-20.8 to 19-20.9. Thank you for pointing it out.

  1. Page 2, line 81-82: why did you divide subjects into two groups (non-obese and overweight/obese), and not into three (underweight, normal weight, overweight/obese)?

Response.

As we mentioned in the introduction and discussion, many investigators and public health professionals have concerned the association between BS and obesity (including overweight), a simple but fundamental question. Then, we would like to answer the question in this study, besides the main outcomes of the association between BS and multiple BMI categories.

  1. Page 3, line 113: it would be better to compare the percentages of underweight women and men

Response.

Thank you for pointing it out. We added the values of percentage (13.9% and 3.0%) in the sentence.

  1. Table 1: I think the superscript c is not correct in the line „Underweight Subjects“

Response.

That is our mistake. We corrected them.

  1. Page 4, lines 139-141: I think, this text should be deleted

Response.

We agree with the comment and deleted them.

  1. Presentation of results: I very much like the presentation of the results in the form of graphs. It makes the results visible and the paper easy to read. However, I think it would be good to provide tables with the results of the logistic regression analyses as supplementary material.

Response.

We agree with the comment and then added the results of logistic regression analyses in a form of table as supplementary material.

  1. Discussion: The possible role of eating disorders should be discussed in more details. It is well-known that eating disorders are associated with breakfast skipping. However, the authors only mention in the last sentence that eating disorders are more prevalent in women than in men.

Response.

We agree with the comment and then added next sentences and three new references in the last of discussion section as follows.

“However, it is unknown whether BS in low-body-weight men is attributable to such eating disorders or impaired mental health, although BS may be associated with depressive symptoms and mood disorders [34-36]. This issue should be elucidated in further study investigating mental health.

  1. Lee YS, Kim TH. Household food insecurity and breakfast skipping: Their association with depressive symptoms. Psychiatry Res. 2019;271:83-88.
  2. Miki T, Eguchi M, Kuwahara K, Kochi T, Akter S, Kashino I, Hu H, Kurotani K, Kabe I, Kawakami N, Nanri A, Mizoue T. Breakfast consumption and the risk of depressive symptoms: The Furukawa Nutrition and Health Study. Psychiatry Res. 2019;273:551-558.
  3. Wilson JE, Blizzard L, Gall SL, Magnussen CG, Oddy WH, Dwyer T, Sanderson K, Venn AJ, Smith KJ. An eating pattern characterised by skipped or delayed breakfast is associated with mood disorders among an Australian adult cohort. Psychol Med. 2019:1-11

Besides reviewers’ comments, We deleted the word of “physician’s” (original Line 238). It is our mistake.

Line 109, The C-statistic, which is equivalent to the area under the curve in the receiver operating characteristic curve, was presented as an index of fitness for each logistic regression model.

Reviewer 3 Report

This study showed that breakfast skipping was associated with a lower BMI in men, particularly in middle aged men. In contrast, BS was associated with a higher BMI in women. Although the study has some value, the manuscript is not okay.  Additionally, a stronger rationale is required to for this manuscript to stand out. There have been multiple studies discussing BS and LND in the past years and you need to show what this study adds to the literature (I suggest you focus on gender differences). Further comments below:

Major comments:

  • What is your rationale for this BMI stratification? (16.9, 17–18.9, 19–20.9, 21–22.9, 23–24.9, 25–26.9, 27–28.9, 29–30.9, 31–32.9, and ≥ 33.0 kg/m2). Based on what you think that BMI of 23 and 21 are different? In terms of disease risk, there is no significant difference. Sticking to the BMI classification might be more acceptable here.  Same applies to WC.
  • You need to clarify what middle age refers to.
  • Abstract: The rationale needs to be improved.
  • Introduction is limited. This topic has attracted a wide interest and therefore should be expanded to include details of previous studies leading to your topic.
  • Discussion shows a limited critical appraisal and is poorly written.

Minor comments:

Title:  I suggest you rephrase it and avoid using the word “unexpectedly”

Keywords: Low body weight: remove low

Round 2

Reviewer 1 Report

The authors have agreed to all the comments provided earlier and have not given any satisfactory answers